# Assessing Amounts of Genetic Variability in Key Horticultural Traits Underlying Core Korean Breeding Lines of Cut Chrysanthemums

**DOI:** 10.3390/plants13050577

**Published:** 2024-02-20

**Authors:** Seung-Young Kim, Haying-Youl Lee, Chanrim Park, Daegwan Kim, Jung-Bun Kim, So-Hyun Kim, Kyeong-Jin Jeong, Ha-Seung Pak, Jae-A Jung, Tae-Sung Kim

**Affiliations:** 1Department of Agricultural and Life Sciences, Korea National Open University, Seoul 03087, Republic of Korea; morra2468@naver.com (S.-Y.K.); lhy1559@hanmail.net (H.-Y.L.); kimjb0712@daum.net (J.-B.K.); rla0417@naver.com (S.-H.K.); 2Department of Bioinformatics & Statisitics, College of Natural Sciences, Korea National Open University, Seoul 03087, Republic of Korea; chanrim228@knou.ac.kr; 3DNACARE, Seoul 06126, Republic of Korea; gardener@dnacare.co.kr; 4Flower Research Institute, Gyeongsangnam-do Agricultural Research & Extension Services, Changwon 51126, Republic of Korea; todtn2019@korea.kr; 5Flower Research Institute, Chungcheongnam-do Agricultural Research & Extension Services, Yesan 32425, Republic of Korea; phsmum@korea.kr; 6Floriculture Research Division, National Institute of Horticultural and Herbal Science, Wanju-gun 55365, Republic of Korea; jabisung@korea.kr

**Keywords:** cut chrysanthemum, standard type, spray type, breeding lines, horticultural traits, genotypic coefficients of variation, phenotypic coefficients of variation, broad-sense heritability, breeding strategy

## Abstract

The cut chrysanthemum holds one of the most substantial segments of the global floriculture market, particularly in Korea. We conducted a detailed assessment of the genetic structures across the cut chrysanthemum breeding lines in Korea. Using standard and spray chrysanthemum breeding lines from leading Korean research institutes, we first compared the variability of 12 horticultural traits, revealing a wide range of variation for most traits. We found that the overall flower diameter (OFD) and ray floret length (RFL) showed a solid positive relationship, regardless of the type. From a multivariate approach, OFD, RFL, and ray floret width (RFW) show consistently high association. Genotypic and phenotypic coefficients of variation analyses further indicated the significant genetic control over most traits. However, certain traits, like the volume of flowers (VF) in standard types, are more influenced by environments. Lastly, our analysis demonstrated substantial variability in broad-sense heritability (*H*); plant height (PH) consistently showed high *H* in both types. But the number of side branches (NOSB) and VF exhibited inconsistent *H* scores. These findings highlight the need for type-specific breeding strategies and modulating environmental management to optimize the trait expressions depending on the *H* scores, which offers significant implications for future breeding strategies.

## 1. Introduction

Chrysanthemum (*Chrysanthemum x grandiflorum* Tzvelev.) is a perennial herbaceous plant belonging the Asteraceae family [1,2]. Cut chrysanthemums, renowned for their diverse forms and vibrant colors, are among the most economically valuable floricultural crops in South Korea and the global market [1,2,3]. These flowers are categorized into two main types: standard and spray chrysanthemums [1]. Standard chrysanthemums are known for one large bloom per stem [2,4]. They are valued for their impressive flower size, long and sturdy stems, and diverse forms [2,4]. On the other hand, spray chrysanthemums are characterized by multiple smaller blooms on each branched stem that create volume and texture in mixed bouquets or vases [1,2,3]. Indeed, since 2018, the total sales volume of the chrysanthemum cut flower has surpassed the rose, and the gap has been gradually more significant [5].

However, the Korean chrysanthemum flower industry, which significantly relies on exports to the Japanese market, has struggled since 2010 [5]. Its market share plummeted due to difficulty in quality control for summer chrysanthemum production, such as deterioration in the plant and flower morphologies, exacerbated by global climate change [5]. This decline in quality has enabled Asian market competitors, such as China and Vietnam, to expand their presence in the Japanese flower market [5]. Furthermore, climate and international economic shifts have worsened the domestic production infrastructure, which is yet to be restored [5]. Establishing effective breeding programs is critical to overcome these imminent challenges [1,5]. The success of these programs fundamentally depends on breeders’ ability to grasp the genetic influence on relevant key traits within the available breeding resources, which prioritizes breeding objectives and devises adequate strategies. But the complex genome structure causes many complications in chrysanthemum breeding programs [4,6,7,8].

The modern cultivated chrysanthemum, a hexaploid with aneuploidy, exhibits a notable genetic complexity, whereby significant variability exists in its chromosome count, comprising approximately 54 chromosomes (2n = 6x = 54) [4,7,8]. This complexity is further accentuated by self-incompatibility, leading to F1 progeny with a wide and distinct range of phenotypic variations, particularly when bred from parents with contrasting traits [1]. The chrysanthemum breeding strategy often results in superior F1 candidates being selected for cultivar development [1].

Genetic complexity is indeed the foundation of the chrysanthemum’s high ornamental value, which is evident in its stunning flowers and vibrant colors across various floral types [6,7,8]. However, this same complexity poses significant challenges for breeders aiming to enhance overall performances systematically.

Breeding efforts often focus on the elegant appearance of chrysanthemum flowers and related attributes [2,3,4,6]. Yet, primary objectives should include improving plant and floral architectures to ensure stability in diverse environments [2,3,4,6]. These enhancements might encompass consistently high plant height, large or voluminous flower sizes, and earlier flowering, regardless of seasonal weather conditions [2,3,4,6]. Such advancements enhance the overall performance of chrysanthemum varieties as crops, further elevating the flowers’ noble appearance, particularly under the current changing climate conditions.

Understanding the broad-sense heritability (*H*) of key traits becomes crucial for chrysanthemum breeders [9,10,11,12,13,14,15,16,17]. *H* provides estimates of all types of heritable genetic variance affecting the expression of a quantitative trait, regardless of the complexity of the genome, as seen in chrysanthemums [9,10,11,12,13,14,15,16,17]. This knowledge offers clear guidance and prioritization in the selection process within the breeding program.

Our study aims to assess the relationship between various traits, the extent of genetic variability that affects phenotypic variation, and the heritability for 12 key horticultural and floral traits in cut chrysanthemum core breeding lines used in Korea. Our findings will provide valuable insights for developing effective and targeted breeding strategies for cut chrysanthemums, which can benefit public- and private-sector breeding programs worldwide, including Korea.

## 2. Results

### 2.1. Phenotypic Variation

We first compared the phenotypic distribution of the 12 horticultural traits between the standard and spray lines (Figure 1, Appendix A); a wide range of variation was observed for most of the studied traits (Figure 2, Appendix A). Different ornamental attributes between the two types caused different variability and population mean accordingly (Figure 2, Appendix A).

In particular, the number of side branches (NOSB), NOF, and volume of flowers (VF) in the standard type show a significantly narrowed phenotypic range compared to the spray-type lines, possibly resulting from an intensive selection process for these traits in the standard chrysanthemum breeding program. In contrast, SD, RFW, and DTF are not significantly different, indicating that the ideal characteristics for the traits share common aspects between the two types (Figure 2, Appendix A).

### 2.2. Trait Correlations

We then investigated the extent of pairwise correlations between the 12 morphological traits (Figure 3 and Appendix A). In most cases, the correlations between the phenotypes are not strong (r < 0.4) (Figure 3 and Appendix A). However, OFD and RFL showed a solid positive relationship regardless of the type (r_standard_ = 0.935, r_spray_ = 0.921), suggesting flower size may be closely dependent on ray floret length (Figure 3 and Appendix A).

We also found a mild but significant positive relationship in PH vs. SD (r_standard_ = 0.537, r_spray_ = 0.705), NOSB vs. NOF (r_standard_ = 0.739, r_spray_ = 0.579), and FSD vs. OFD (r_standard_ = 0.490, r_spray_ = 0.639) combinations. We noticed that the correlation of PH vs. SD and FSD vs. OFD in the spray is much stronger than the standard types, which would be useful in the spray breeding programs (Figure 3 and Appendix A).

We performed a multivariate analysis using 12 traits evaluated in 2021 and 2022 to understand trait associations comprehensively (Figure 4 and Appendix A). PCA revealed that the first two principal components (PCs) explained almost half of the total variation in the data set, accounting for 50.1%, 44.9%, and 50.3% of the variability in the cut, standard, and spray chrysanthemum types, respectively (Figure 4 and Appendix A). We conducted a factor analysis to investigate possible connections among phenotypes within a principal component or a latent factor (Appendix A).

In the cut chrysanthemum with 31 standard and 43 spray breeding lines, factor 1, which accounts for 27% of the total variation, is associated with OFD, RFL, FSD, and RFW. OFD and RFL are strongly linked with factor 1, with factor loadings of 0.97 and 0.92, respectively (Appendix A). Contrastingly, FSD and RFW are only marginally connected to factor 1, with factor loadings of 0.57 and 0.50 (Appendix A). Factor 2, responsible for 23% of the total variability, includes NOF, NOSB, SD, and PH with factor loadings of 0.79, 0.73, 0.65, and 0.61, respectively (Appendix A). In the standard types, factor 1 contributes 29.7% of the total variation, wherein OFD (0.91) and RFL (0.87) are strongly linked, while RFW (0.5) is weakly associated with it. Factor 2 explains 15.2% of the total variability and includes NOF (0.71), NOSB (0.65), SD (0.64), and PH (0.51) in the association (Appendix A).

In the spray type, factor 1 and factor 2 account for 32.8% and 17.6% of total variability, respectively (Appendix A). RFL, OFD, and RFW are grouped in factor 1 with the factor loading of 0.93, 0.88, and 0.62, whereas VF (0.99) and DFD (−0.76) are relatively strong but negatively correlated within factor 2 (Appendix A). The results support that RFW could be linked to OFD and RFL in factor 1, irrespective of the floral types, implying that RFL predominantly influences OFD, while RFW may have a minor impact on OFD (Appendix A).

Unlike factor 1, factor 2 displays distinct associations exclusive to its type (Appendix A): In both the cut and standard types, firm connections are observed between PH vs. SD and NOSB vs. NOF, which are clustered together in factor 2, as shown in Figure 3 (Appendix A). However, this particular pattern of association is absent in the spray type (Appendix A), possibly indicating the underpinning regulatory mechanism might vary among traits between the floral types.

### 2.3. Genetic and Phenotypic Coefficients (GCV and PCV)

We calculated the GCV and PCV to determine the genetic contribution to the variation of a trait (Table 1). We specified GCV and PCV for the different types to avoid over- or underestimation of the genetic contribution to trait variation, as we discovered type-specific differences in certain traits between the two types.

Our analysis revealed that the PCV values of the standard type varied between 8.69 and 95.51 (Table 1). Among them, DFD had the highest PCV, while NOF, NOSB, FSL, RFW, FSD, PH, and SD demonstrated high PCV values (Table 1). RFL, OFD, and DFB had moderate PCV values, while VF had a low PCV (Table 1). Most of the traits showed high GCV values and had a linear relationship with PCVs, suggesting there is significant genetic control and less influence of environmental factors affecting the traits’ variability (Appendix A). However, the GCV of VF was low, along with low PCV, indicating limited genetic variability mediated by narrow overall variation within the population (Table 1).

The PCV values in the spray type are largely similar to those in the standard type, ranging from 13.04 to 77.57 (Table 1); DFD had the highest PCV value. NOSB, NOF, FSL, VF, RFW, RFL, OFD, FSD, PH, and SD showed still relatively high PCV values, while DTF had moderate PCV values (Table 1). As in the standard type, most traits exhibited high GCV values and correlated linearly with PCVs (Appendix A). However, NOSB and NOF had relatively lower GCV values, deviating from the linear relationship, indicating lower genetic influences on the variability of NOSB and NOF traits in the spray type (Appendix A).

We found that the linear model strongly explains the relationships of PCVs between the standard and spray lines; about 70% of the dataset follows this model, indicating that shared traits are integral to the phenotypic selection strategies in cut chrysanthemum flower breeding (Appendix A). However, spray chrysanthemums exhibit higher VF values, while standard types tend to have lower VF. These variations may be attributed to uniqueness in the initial breeding materials or distinct phenotypic selection criteria in the respective breeding program (Appendix A). Moreover, the linear model barely explains the relationship of GCVs between standard and spray, where only 33.7% of the dataset conforms to the model, indicating that genetic contribution to the phenotypic variability between the two types may differ (Appendix A).

### 2.4. Heritability

High GCVs relative to PCVs exist across all traits studied. We calculated the *H* of the traits in the two chrysanthemum types to estimate how much of the observed genetic variation can be inherited (Table 1, Figure 5). The average heritability value for the standard type of chrysanthemum is 0.64, with values ranging from 0.4 to 0.88 (Table 1, Figure 5). We observed that the VF trait of the standard chrysanthemums exhibits the lowest heritability (0.4), suggesting a considerable environmental influence on this trait (Table 1, Figure 5). The NOF and NOSB traits, with heritability values of 0.37 and 0.47, have the second lowest heritability (Table 1, Figure 5). The similarity in heritability between the traits aligns with their correlated nature. The traits, including OFD, FSD, DFB, SD, DFD, and RFL, fall under average (Table 1, Figure 5). In contrast, RFW, PH, and FSL show significantly higher heritability, indicating a more substantial genetic contribution in their phenotypic variation (Table 1, Figure 5).

It was found that similar patterns in the average and range of H scores (mean = 0.65, range (0.06–0.92)) in the spray type (Table 1, Figure 5). Interestingly, the NOSB and NOF traits had very low heritability values of 0.06 and 0.16, respectively, indicating that environmental factors may greatly impact the variability of these traits (Table 1, Figure 5). The similar heritability between the two traits, as observed in standard types, is likely due to their correlated nature, possibly a correlation in their genetic control. However, lower heritability may be due to less selective breeding pressure for these traits in spray chrysanthemums, a factor worth considering in future breeding programs. The traits DFB, FSD, SD, RFW, RFL, and FSL fall within the average range, while PH, DFD, OFD, and VF surpass the average heritability threshold (Table 1, Figure 5).

Many traits, including PH, SD, FSL, FSD, DFD, RFL, RFW, and DTF, demonstrated relatively high heritability in both standard and spray chrysanthemum types (Table 1, Figure 5). However, certain traits, like OFD, NOF, NOSB, and VF, are quite different between the two types. As a result, no significant linear relationship was found between the two types in terms of heritability scores, suggesting that the genetic determinants of certain traits may differ between standard and spray chrysanthemums, pointing to a complex and type-specific genetic architecture (Appendix A).

## 3. Discussion

In this study, we evaluated the relationship between various traits, the extent of genetic variability that affects phenotypic variation, and the heritability for 12 key horticultural and floral traits in cut chrysanthemum core breeding lines used in Korea. Our findings reveal relatively high *H* scores across many key horticultural traits in general but notable differences in *H* scores in certain traits between standard and spray chrysanthemum types, providing valuable insights for future breeding strategies to reach market competitiveness.

To strengthen our arguments, we selected over 30 lines of standard-type and spray-type chrysanthemums, originating from leading chrysanthemum research institutes in South Korea, with various flower attributes within each type covering a broad genetic spectrum (Appendix A). Our tightly controlled methodical approach in plant preparation and phenotypic evaluation provided a comprehensive and reliable dataset. In addition, the balanced ANOVA approach—considering genotypes, block, and year effects as a random factor—was pivotal in accurately estimating the variance components, allowing for relative comparison of *H* across the traits. Future studies that incorporate location replications will refine the estimation.

### 3.1. The Correlation and the Multivariate Analyses to Explore Trait Association between the Standard and Spray Chrysanthemum Lines

Our study revealed various degrees of correlation among the 12 horticultural traits in chrysanthemums. The strong positive correlation between overall flower diameter (OFD) and ray floret length (RFL) in both standard and spray types is particularly notable, suggesting that RFL may be a key determinant of OFD in the cut chrysanthemums. However, the correlation between plant height (PH) and stem diameter (SD), as well as between flower stem diameter (FSD) and OFD, varied significantly between the standard and spray types, indicating that while certain traits may be genetically linked in one type, the same might not be valid for the others. Such insights are critical for breeders, highlighting the need for type-specific breeding approaches.

Multivariate analysis, with the PCA based factor analysis, provided a more comprehensive perspective on the relationships among the traits [18]. In the first two PCs, which capture key aspects of the phenotypic variation, we found the complex interplay of traits among RFW, RFL, and OFD in both types through the factor analysis, highlighting their networking role in the phenotypic variation in chrysanthemums. In contrast, the distinct trait associations observed within each type point to the unique genetic architecture between standard and spray chrysanthemums, indicating that the genetic basis of particular traits may differ significantly between the two types.

The correlations and multivariate analysis results have profound implications for chrysanthemum breeding. The strong correlation between OFD and RFL across types can be key targets for breeders focusing on flower size and shape. Since these traits are likely genetically linked, selecting one could improve the other, simplifying the breeding process. However, the differences in trait correlations between standard and spray types indicate the necessity of type-specific breeding strategies. Understanding these common and unique genetic architectures allows breeders to tailor their approaches by leveraging the genetic correlations or the distinct genetic architectures of standard and spray chrysanthemums. This approach has already been effectively used to improve ornamental value and market competitiveness, and it continues to do so.

### 3.2. GCV and PCV among the Standard and Spray Chrysanthemum Lines

Our analysis revealed a broad range of GCV and PCV values across different traits in both standard and spray types of chrysanthemums. High GCV values observed in traits like FSL and FSD indicate substantial genetic control. This finding suggests these traits are highly amenable to genetic improvement [19,20].

In contrast, traits such as volume of flower (VF) showed lower GCV and PCV values, particularly in standard chrysanthemum types. This pattern implies that VF is less influenced by genetic factors and more by environmental conditions [13,19,21]. Given VF’s importance in standard chrysanthemums [1,2], the observed limited genetic variability could be attributed to restricted genetic resources in the breeding pool. Therefore, future breeding programs aiming to enhance VF in standard chrysanthemums should focus on two efforts: firstly, expanding the genetic resources to introduce more variability, and secondly, refining environmental management practices to optimize VF expression. Such a dual approach will enable more consistent expression of VF, ensuring its stability and quality under various environmental conditions.

We found that PCVs between standard and spray types more tightly followed the linear model than GCVs in the data set, suggesting that some underlying genetic controls could be distinct, although phenotypic traits may appear similar. This insight is crucial for breeders, implying that breeding strategies effective for the standard types may not be directly transferable to the sprays, and vice versa, in certain traits, including VF, NOSB, and NOF.3.3. 

Broad-Sense Heritability in Chrysanthemum Traits.

Our study’s broad-sense heritability (*H*) values varied significantly across different traits, reflecting the diverse genetic influences on these traits in the cut chrysanthemums. Many traits, including PH, SD, FSL, FSD, DFD, RFL, RFW, and DTF, demonstrated consistently high heritability in standard and spray chrysanthemum types. However, there was quite a difference in the *H* score among certain traits, suggesting the *H* scores vary dependently by trait between the types.

Specifically, PH, FSL, and RFW in standard chrysanthemums and PH, DFD, and VF in the spray exhibited significantly high heritability, indicating that these traits are more consistently inherited from parent to offspring and are more responsive to selective breeding efforts. Among the traits, PH consistently showed high heritability, implying its historical importance in cut chrysanthemum breeding programs [19,22]. The relatively high PCV and GCV for PH in both types suggest a robust gene pool influencing this trait, suggesting that PH can be reliably targeted in future Korea (K) standard or spray breeding programs, offering stable selection opportunities [19,22].

On the other hand, traits like NOSB and VF exhibited inconsistent heritability scores between standard and spray chrysanthemum types, implying distinct genetic underpinnings and breeding histories for these traits. The high heritability of VF in spray chrysanthemums versus its lower heritability in standard types may likely reflect a historical focus on maximizing flower volume in spray varieties. This focus would have led to the introduction of diverse but reliable gene pools, which may potentiate a more robust selection of genetic factors improving VF in sprays.

In contrast, standard chrysanthemums might have been selectively bred with priorities, such as larger flower sizes or fewer side branches, successfully improving these aspects of genetic controls in NOSB and OFD respectively, but potentially narrowing the genetic variability of VF, which happens to, coincidently or not, be less stable in the given environment. As such, irrespective of a genetically favorable background, the expression of VF in the standard—as in NOSB of spray background—could be significantly modulated by external factors such as growing conditions and cultivation practices. Therefore, breeding strategies for these traits must also incorporate considerations for broadening robust gene pools or modulating environmental management to optimize the trait expression.

As such, these disparities in heritability scores thus highlight the necessity for type-specific improvement strategies in the K-chrysanthemum breeding program. The breeding programs could effectively focus on enhancing NOSB for standard chrysanthemums, showing higher heritability. Meanwhile, for spray chrysanthemums, breeding efforts might be more fruitfully directed towards traits like VF, capitalizing on its higher genetic influence. However, as a long-term plan, an effort should be made to compensate for the relevant limited genetic resources.

Our findings on heritability in chrysanthemums open several avenues for the K-chrysanthemum breeding program. Another promising future area is the exploration of genetic markers associated with high-heritability traits, which could facilitate more efficient and targeted breeding efforts. The chrysanthemum genomics field has seen significant advancements in recent years, unraveling the complex genome of chrysanthemums [3,7,8]. These studies offer a valuable foundation for further genetic analysis, such as GWAS (genome-wide association study) and QTL (quantitative trait loci) mapping, which can efficiently scan the chrysanthemum genome to locate SNP markers linked to the agriculturally important traits, leveraging the availability of genomic sequence data and high-throughput genotyping technologies [3,7,8,22]. Recently, GWAS was successfully exploited to identify specific genetic variants associated with PH [22], suggesting this genomics approach can be implemented someday for other important traits with high *H* based on our heritability findings.

## 4. Materials and Methods

### 4.1. Plant Materials and Preparation

We obtained 31 standard-type and 41 spray-type chrysanthemum breeding lines or cultivars from representative chrysanthemum research institutes in Korea, including the National Institute of Horticultural and Herbal Science (NIHHS), Flower Breeding Research Institute of Gyeongsangnam-do Agricultural Research & Extension Services (FBRIG_ARES), and Flower Research Institute of Chungcheongnam-do Agricultural Research & Extension Services (FRIC_ARES), which includes various flower shapes, and their detailed specification is listed in Appendix A.

We induced the root simultaneously from leaf cuttings of all 72 chrysanthemum lines and raised the rooted plantlets for one month. In late June, the one-month-old seedlings were transplanted under a plastic house with a 15 cm × 15 cm density in the artificial soil bed. Each line was allocated with a randomized complete block design consisting of five replications for observation in two independent beds. The soil bed included peat moss and perlite in equal ratios(*v*/*v*). All the horticultural practices for the chrysanthemum cultivation, including irrigation and fertigation, were applied uniformly to the beds during the cultivation period, including short-day treatment. For the short-day condition, the diurnal state with 11 h of light and 13 h of darkness was implemented about eight weeks after transplanting.

### 4.2. Phenotypic Evaluation and Data Analysis

Phenotypic evaluations were performed on the chrysanthemum lines that showed 50% blooming after early October in 2021 and 2022. We investigated the following 12 horticultural traits from selected plants relevant to the floricultural qualities, including (1) plant height (cm), (2) stem diameter (mm), (3) number of side branches (each), (4) number of flowers (each), (5) flower stem length (mm), (6) flower stem diameter (mm), (7) overall flower diameter (mm), (8) flower center diameter (mm), (9) number of petals (1–6 scale), (10) petal length (mm), (11) petal width (mm), and (12) days to flowering. Detailed information on the evaluation is shown in Figure 1 and described in Appendix A.

Pearson correlation coefficients between the mean phenotypic traits were calculated to measure the strength of linear association between the traits. We further evaluated the strength of the associations among the traits on a multivariate scale based on principal component analysis (PCA)-based factor analysis. PCA was conducted to reduce the dataset’s dimensionality to two principal components. Subsequently, a varimax rotation was applied to these components, clarifying the contribution of each trait to the principal components. To ascertain the most influential trait combinations, we established a loading cutoff of 0.6. Traits that exhibited loadings equal to or exceeding this threshold were thought to have a strong association with the respective principal components. The JMP16 was implemented for these statistical analyses.

### 4.3. Estimation of the Heritability

The PCV and GCV were estimated as previously described [19,21]. The GCV and PCV estimates were classified as follows [19,20,21]: a value of 0–4% is represented as low, 5–14% is moderate, 15–30% is high, and >20% is very high.

We dissected the relative genetic contribution from the non-genetic effect underlying phenotypic variation, implementing the *H* score as previously implemented [19,20,23]. The relevant variance components were estimated from the balanced ANOVA, considering that genotype, block, and year effects are random. As a result, the *H* of the 12 horticultural traits by various floral types were calculated using the ‘sommer’ package in R [24], classifying them in low (below 30%), medium (30–60%), and high (above 60%) levels [19,20,23].

## 5. Conclusions

Our study, including the *H* study, offers valuable insights into the possible amount of genetic control of key horticultural traits in chrysanthemums. These findings have significant implications for breeding strategies, highlighting the potential for genetic improvement in traits with high heritability and the need for more complex approaches for traits that are more significantly influenced by environmental factors. By leveraging these insights, breeders can develop more effective and integrated strategies to genetically enhance chrysanthemum varieties, ultimately advancing the K-chrysanthemum industry.

## Figures and Tables

**Figure 1 plants-13-00577-f001:**
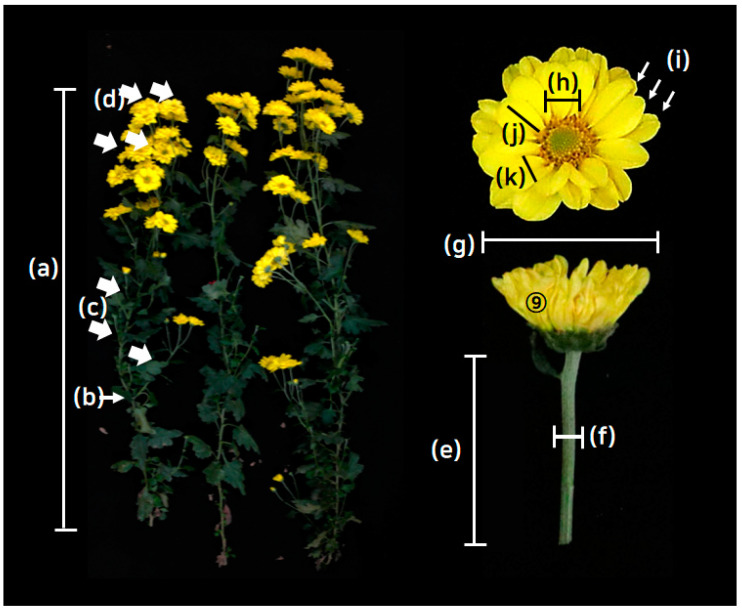
Visual description of the phenotypic evaluations for the horticultural traits of chrysanthemum breeding lines used in this study. (a) Plant height (PH, cm), (b) stem diameter (SD, mm), (c) no. of side branches (NOSB, each), (d) no. of flowers (NOF, each), (e) flower stem length (FSL, mm), (f) flower stem diameter (FSD, mm), (g) overall flower diameter (FD, mm), (h) disk flower diameter (DFD, mm), (i) volume of flower (VF, 1~6 scales), (j) ray floret length (RFL, mm), (k) ray floret width (PW, mm). The unit and abbreviation of each phenotypic assessment is indicated in the parenthesis in a–k. The chrysanthemum line shown in the figure is ‘Yes holic’, as indicated in no. 64 of Appendix A. The arrows indicate the specific areas or positions for the phenotypic evaluations. Detailed methods are described in Appendix A.

**Figure 2 plants-13-00577-f002:**
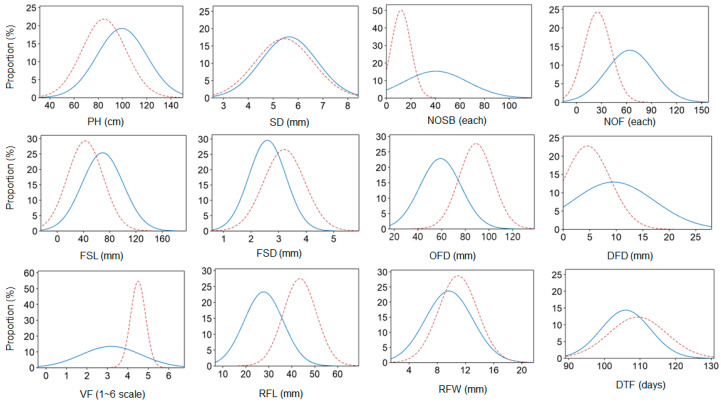
Density curve of the selected horticultural traits for standard and spray chrysanthemum breeding accessions. The original histogram for either the standard (blue solid line) or spray (red dotted line) type was fit to a normal distribution, utilizing data from the 2-year overall dataset. Y axis represents an estimated percentage of a trait value within a group. PH stands for plant height; SD, stem diameter; NOSB, no. of side branch; NOF, no. of flowers; FSL, flower stem length; FSD, flower stem diameter; OFD, overall flower diameter; DFD, disk flower diameter; VF, volume of flower; RFL, ray floret length; RFW, ray floret width; DTF, days to flowering. The unit of each phenotype is indicated in the parenthesis side by the abbreviation on the *x*-axis. The methods of the phenotypic assessment were performed as described elsewhere.

**Figure 3 plants-13-00577-f003:**
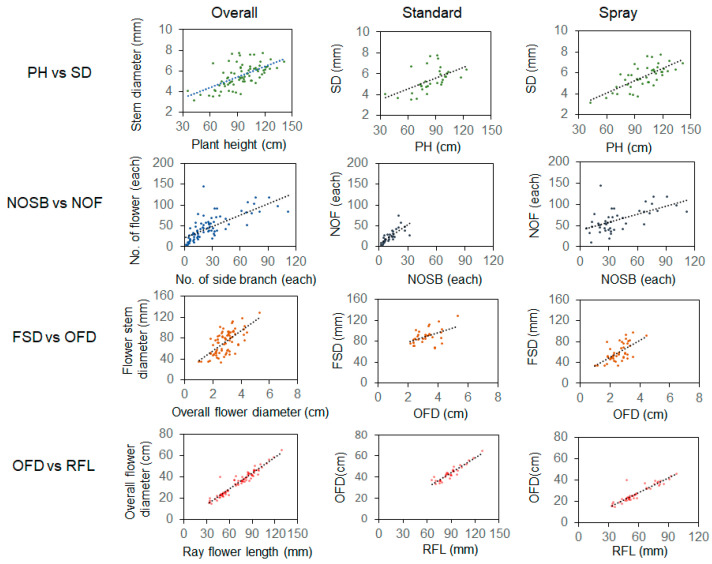
Pairwise correlations between horticultural traits in chrysanthemum breeding lines. This result delineates the pairwise correlations between various horticultural traits across three chrysanthemum breeding line types: cut, standard, and spray. It emphasizes the trait combinations in the cut type that exhibit a Pearson’s correlation coefficient value exceeding 0.6, which is further extended to the standard and spray types, enabling a comparative examination of the consistency and variability of these correlations across the different types. PH stands for plant height; SD, stem diameter; NOSB, no. of side branches; NOF, no. of flowers; FSD, flower stem diameter; OFD, overall flower diameter; RFL, ray floret length.

**Figure 4 plants-13-00577-f004:**
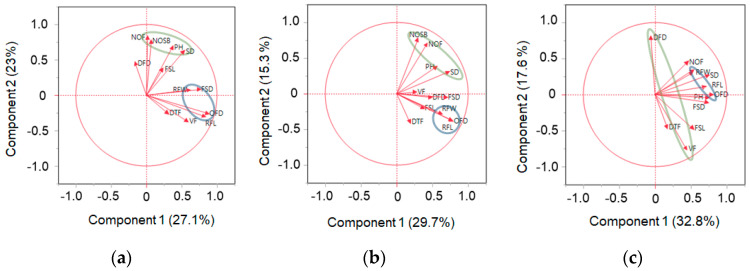
PCA loading plot of 12 horticultural traits in the cut, standard, and spray chrysanthemum breeding lines. This figure illustrates a graphical summary of principal component analysis (PCA) results based on the 2-year data of the trait values in the cut (**a**), standard (**b**), and spray (**c**) chrysanthemum breeding lines of our study. Each loading plot depicts the interrelationships among the traits within each type, explained by Principal Component 1 (PC1, *x*-axis) and PC 2 (*y*-axis). The traits within blue or green circles are the ones that show strong associations in factor 1 and factor 2, respectively. PH stands for plant height; SD, stem diameter; NOSB, no. of side branch; NOF, no. of flowers; FSL, flower stem length; FSD, flower stem diameter; OFD, overall flower diameter; DFD, disk flower diameter; VF, volume of flower; RFL, ray floret length; RFW, ray floret width; DTF, days to flowering.

**Figure 5 plants-13-00577-f005:**
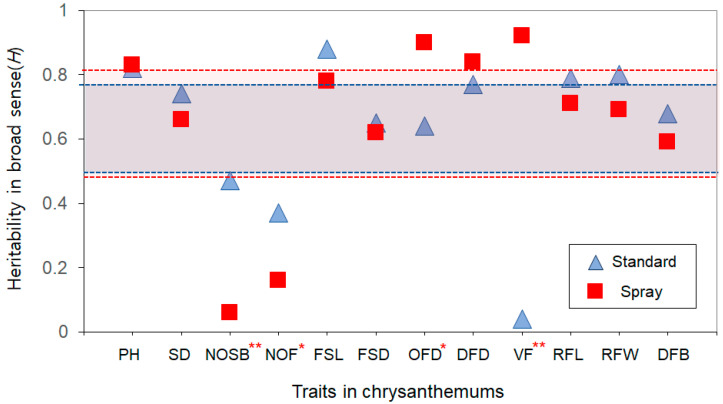
Heritability distributions for the 12 horticultural traits in the standard and spray chrysanthemum breeding lines. This figure illustrates the mean broad-sense heritability (*H*) scores in standard (blue triangles) and spray (red rectangles) chrysanthemum lines. The blue and red dotted lines delineate the confidence intervals for the heritability estimates in the standard and spray types, respectively. Statistical annotations next to a trait are used to highlight significant deviations in the *H* score between the two types for each trait: ** denotes a deviation exceeding two standard deviations in the *H* score between standard and spray types, indicating a substantial difference in the *H* between the two types for a specific trait. * represents a deviation that exceeds one standard deviation but is less than two. The standard deviation calculations are based on all the mean *H* values across the overall traits in both types. PH stands for plant height; SD, stem diameter; NOSB, no. of side branch; NOF, no. of flowers; FSL, flower stem length; FSD, flower stem diameter; OFD, overall flower diameter; DFD, disk flower diameter; VF, volume of flower; RFL, ray floret length; RFW, ray floret width; DTF, days to flowering.

**Table 1 plants-13-00577-t001:** The estimation of broad-sense heritability among the horticultural traits of the standard and spray chrysanthemum lines of the study.

Traits	GCV ^z^ (%)		PCV ^y^ (%)		*H* ^x^	
	Standard	Spray	Standard	Spray	Standard	Spray
PH	19.70	19.44	22.88	21.26	0.82	0.83
SD	19.63	16.89	22.71	20.73	0.74	0.66
NOSB	47.78	15.73	68.06	63.22	0.47	0.06
NOF	42.79	19.65	69.95	48.27	0.37	0.16
FSL	62.01	40.82	65.74	46.17	0.88	0.78
FSD	19.01	21.09	24.26	26.59	0.65	0.62
OFD	13.55	27.78	16.97	29.22	0.64	0.9
DFD	83.86	71.11	95.51	77.57	0.77	0.84
VF	1.94	44.31	8.69	45.99	0.04	0.92
RFL	15.26	25.54	17.19	30.2	0.79	0.71
RFW	23.05	28.87	25.69	34.53	0.80	0.69
DTF	12.80	9.98	15.56	13.04	0.68	0.59
Mean	30.12	28.43	37.77	38.07	0.64	0.65
Upper 95% ^w^	45.39	39.07	55.98	50.06	0.79	0.82
Lower 95% ^u^	14.84	17.80	19.55	26.07	0.49	0.47

^u^, lower 95% refer to lower limit of the confidence interval of mean; ^w^, upper 95% refer to upper limit of the confidence interval of mean; ^x^, H is broad-sense heritability; ^y^, PCV is phenotypic coefficient of variation; ^z^, GCV is genotypic coefficient of variation; PH stands for plant height; SD, stem diameter; NOSB, no. of side branch; NOF, no. of flowers; FSL, flower stem length; FSD, flower stem diameter; OFD, overall flower diameter; DFD, disk flower diameter; VF, volume of flower; RFL, ray floret length; RFW, ray floret width; DTF, days to flowering.

## Data Availability

Data will be available upon reasonable request to authors.

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
