# Peer review of "Assessing Amounts of Genetic Variability in Key Horticultural Traits Underlying Core Korean Breeding Lines of Cut Chrysanthemums"

_plants, 2024, doi:10.3390/plants13050577_

Round 1
Reviewer 1 Report
Comments and Suggestions for Authors
The work presented by Seung Young Kim, Haying Youl Lee et al. focus on the study of genetic variability influencing phenotypic variation, and the heritability of 12 horticultural and floral traits of chrysanthemum. The topic of this research is of a great importance, and all these results can help in new breeding programs and ongoing breeding works, because authors tested 75 different Chrysanthemum breeding lines and cultivars. All activities related to research of different genotypes of every plant species result in the characterization of a variety of economically important quantitative and qualitative traits. These studies and knowledges are required for the further practical use of cultivars in plant breeding, overcome the developed barriers due to the climate change.
The structure of the manuscript follows the requirement of Plants.
General comments:
1. Very comprehensive, detailed analyses and evaluations were presented in the manuscript.
2. ‘Introduction’ and ‘Material and methods’ parts need improvement, including minor English language editing.
Introduction
Line 38. Is this the correct scientific name of chrysanthemum: Chrysanthemum morifolium? Please, check!
Line 38. Asteraceae (italic)
Line 54-55. “However, the complex genome structure of the plants causes” instead of “their intricate genome structure adds”
Line 57-73. Please, describe logically and more precisely the difficulty of the chrysanthemum breeding!
Line 74. Please, paraphrase and rewrite the concrete aim of the study!
I have two questions:
1. What is the most important advantage of (growing) chrysanthemum in general?
2. What is the effect of climate change on chrysanthemum production; and its qualitative and quantitative traits?
Please, consider inserting the answers in the ‘Introduction’
Material and methods
Line 385. „We took roots from leaf cuttings…” means „Root growing were induced…”?
Line 387. „green house” instead of „plastic house”
In „4.2. Phenotypic evaluation and data analysis” part (Line 403-412.) please, be a little more precise when describing the methods!
Result and discussion
These are well written parts and including valuable results and information.
References
Number 2. and 4. references are the same!
Su, J.; Jiang, J.; Zhang, F.; Liu, Y.; Ding, L.; Chen, S.; Chen, F. Current achievements and future prospects in the genetic breeding of chrysanthemum: a review. Horticulture Research 2019, 6, 1-19, DOI 10.1038/s41438-019-0193-8.
Please, make all the necessary corrections! I suggest checking all the references!
Comments on the Quality of English LanguageIn the Introduction and Material methods part minor English editing is needed.
Author Response
Dear Reviewer1,
Thank you for allowing us to submit a revised draft of the manuscript "Assessing Amounts of Genetic Variability in Key Horticultural Traits Underlying Core Korean Breeding Lines of Cut Chrysanthemums" for publication in plants. We appreciate the time and effort spent on the reviewing process and the dedicated comments, which would improve our paper. We have incorporated most of the suggestions made by the reviewer 1, and those changes are in 'blue' within the manuscript. Please see the attached word document in blue, for a point-by-point response to the reviewers' comments and concerns. All page numbers refer to the revised manuscript file.

Reviewer 2 Report
Comments and Suggestions for Authors
My comments are all in the edited MS.

minor edits
Author Response
Thank you for allowing us to submit a revised draft of the manuscript "Assessing Amounts of Genetic Variability in Key Horticultural Traits Underlying Core Korean Breeding Lines of Cut Chrysanthemums" for publication in plants. We appreciate the time and effort spent on the reviewing process and the dedicated comments, which would improve our paper. We have incorporated most of the suggestions made by the reviewer2, and those changes are in 'blue' within the manuscript. Please see the attached word document file in blue, for a point-by-point response to the reviewers' comments and concerns. All page numbers refer to the revised manuscript file.
